# Coping Styles in Patients with Parkinson’s Disease: Consideration in the Co-Designing of Integrated Care Concepts

**DOI:** 10.3390/jpm12060921

**Published:** 2022-06-01

**Authors:** Johanne Stümpel, Marlena van Munster, Sylvie Grosjean, David J. Pedrosa, Tiago A. Mestre

**Affiliations:** 1Cologne Center for Ethics, Rights, Economics, and Social Sciences of Health (Ceres), University of Cologne, 50923 Cologne, Germany; 2Center for Life Ethics, University of Bonn, 53113 Bonn, Germany; 3Department of Neurology, University Hospital Marburg, 35037 Marburg, Germany; marlena.vanmunster@uni-marburg.de (M.v.M.); david.pedrosa@uni-marburg.de (D.J.P.); 4Department of International Health, CAPHRI Care and Public Health Research Institute, Maastricht University, 6200 MD Maastricht, The Netherlands; 5Department of Communication, University of Ottawa, Ottawa, ON K1N 6N5, Canada; sylvie.grosjean@uottawa.ca; 6Center of Mind, Brain and Behavior, Philipps-University Marburg, 35032 Marburg, Germany; 7Parkinson Disease and Movement Disorders Centre, Division of Neurology, Department of Medicine, The Ottawa Hospital Research Institute, University of Ottawa Brain and Mind Research Institute, Ottawa, ON K1Y 4E9, Canada; tmestre@toh.ca

**Keywords:** Parkinson’s disease, integrated care, personalized care, multidisciplinary care, coping styles, personality traits

## Abstract

Integrated care models may help in designing care for Parkinson’s disease (PD) that is more efficient and patient-centered. However, in order to implement such models successfully, it is important to design these models around patients’ needs and preferences. Personality traits and coping styles play a well-studied important role in patients’ disease perception and their utilization of medical and social services to cope with their disease. There is evidence that coping styles remain largely unchanged over the course of PD; coping styles are defined in the early stages of life and extend over the entire lifespan of the patient. Therefore, it seems necessary to consider aspects of the personality traits and coping styles of PD patients in the development and implementation of care models. We postulate that by taking patients’ personality traits and coping styles into account, care models for PD can be designed in a more individualized and, thus, more effective way. This paper, structured in three main sections, attempts to structure the uptake of patients’ coping styles in the co-design of integrated care models. However, further studies are needed to better develop tailored care concepts to the needs of people living with PD and their individual coping styles.

## 1. Introduction

Parkinson’s disease (PD) is the second most common neurodegenerative disease after Alzheimer’s disease, and it presents with a wide variety of motor and non-motor symptoms [1]. Neuropsychiatric symptoms, such as depression and anxiety [2,3], are likely to occur long before a PD diagnosis and can negatively impact the quality of life of people living with Parkinson’s (PwPs) and their care partners [4]. Care delivery in PD is of a complex nature, and personalized care concepts might be able to address this complexity [5]. However, such care approaches are not yet widely applied in the field of PD care [6]. Chronic diseases such as PD affect more than 80% of people over 65 in the European Union, posing an immense challenge to health and social systems in those countries [7]. It is expected that healthcare resources may become scarce due to an increasing burden of chronic diseases associated with an aging population, such as PD [8,9]. Therefore, a shift towards more effective care concepts is needed [10,11,12]. Integrated care concepts have been proposed as potential solutions, aiming to improve the quality and availability of care [6,11,12,13,14]. Integrated care is a complex construct and can be defined in different ways [15]. For this publication, we consider the concept of integrated care based on the work of Goodwin and Stein [16]:(1)A required response to overcome fragmentation in care delivery.(2)An approach to improve care quality and its cost-effectiveness.(3)A service innovation guided by the principle of people- and population-centeredness.

In order to master the transformation of care fragmentation towards integrated care approaches, it is essential to involve patients more closely in the design of healthcare [17].

The project iCare-PD [18] aims to address this need by developing a technology-supported, home-based, and community-centered integrated care model. A central pillar of the project is that patients are involved in the design process [17,19]. As part of this so-called co-design process, we conducted qualitative interviews with PwPs and their care partners and concluded that they were coping and organizing their lives with PD in very different ways [20]. Some PwPs and their care partners self-reported their wish to be an integral part of the decision-making process of their medical and social care at any time. However, others did not want to be involved. Studies in other areas show that the consideration of non-diagnostic patient factors, such as coping styles, are important considerations in the design of effective therapies [21,22,23]. Therefore, it seems important to discuss the consideration of coping styles in the design of healthcare for PD patients.

In this paper, we outline the potential value of coping styles in the co-design of an integrated care model for PD as means to improve care delivery and present the approaches to successfully achieve it.

## 2. Co-Design in Integrated Care Concepts: Research with and Not on PwPs

According to Goodwin and Stein [15], one approach to improve the quality of care and increase the focus on patients and entire populations is to involve the stakeholders in the design process of new care concepts from the onset. Hence, co-creation, meaning a joint development by researchers and stakeholders, might be usefully employed for this purpose. Co-design approaches originate from the manufacturing industry. Starting in the 1970s, the industry underwent a shift from designing products for people (supplier-centered design) to designing products with people’s needs in mind (user-centered design), for which designers, suppliers, and consumers work together to consider a problem and develop a solution (co-design) [24]. Co-design approaches applied in research may enable the development of healthcare delivery services and healthcare technology directly and in close collaboration with future users [25,26]. In co-design, the user’s perspective and experience are critical to the development process [26], which is why the concepts of collaboration and engagement are central to any co-design approach [27,28]. Co-design approaches link doing, talking, thinking, feeling, and the exchange of experience and knowledge in a complex process [27]. Co-design may also be valuable for improving existing care models [29]. A crucial aspect of any co-design process is the selection of methods suitable for stakeholders (e.g., deciding on a virtual discussion format for healthcare professionals) and the context to ensure that the outcomes, meaning the defined requirements of a care model, are perceived as both realistic and feasible by participants [30,31]. We will start by outlining the experience-based co-design (EBCD) approach as an example of a co-design approach widely applied in the healthcare sector [31]. This approach combines participatory design and user experience design to improve the quality of healthcare services. It was originated in 2005/06 as a participatory action research approach that drew upon design theory [26] and is organized into the following four phases [32,33] in Table 1.

Co-design in PD care is becoming more important and more frequently applied [19,34,35], so an increase in knowledge on the possibility of the operationalization of this approach is to be expected. The use of co-design in PD may result in a shift from a routine, episode-based approach to care to a more flexible approach to care delivery based on the mutual needs of patients, care partners, and care providers [34]. However, Wannheden and Revenäs (2020) suggest that further research is needed to examine the concept of co-design in PD as well as its relevance to self-care and healthcare [34].

At this point, we can conclude that the concept of co-design is not only becoming increasingly important when it comes to the (re-)design of care concepts for the chronically ill but that the collaboration and engagement of all stakeholders (PwPs, care partners, and care providers) are needed in the development process from the very beginning. In this way, the experiences and needs of all those involved in PD care are incorporated from the outset and increase the acceptance of the subsequent users.

Why it is crucial to not only design with certain concepts in mind but to take a closer look at the individuals for whom these concepts are intended when developing care delivery to PwPs will be explained in the next section.

## 3. Addressing the Question of Personality—Take a Closer Look at Our Patients!

Several lines of evidence indicate a strong relationship between personality traits and coping styles. Lazarus and Folkman (1984), among others, highlight that personality cannot be neglected when an individual’s primary appraisal of stressors is investigated [36]. An individual’s primary appraisal, i.e., the way an individual interprets and appraises a stressor, is influenced by personality traits such as neuroticism or extraversion [37], which may impact emotional and behavioral responses to stressors. Subsequently, individuals attempt to cope with the situation by trying to make the stressor manageable. If the individual fails to cope with the stressor, poor emotional outcomes may result, often resulting in anxiety [36]. The personality structure (the so-called personality) is formed from the entirety of personality traits found in a given person [38]. Concepts for structuring these traits can be found in a variety of personality theories and models [39,40]. Personality traits describe relatively consistent —even over time—characteristics of a person that define how they think, feel, and behave in a certain way. Personality traits perceived as positive are generally referred to as strengths and negative ones as weaknesses. Among personality traits, the most widely used model is the so-called five-factor model. Five general personality traits are included in this model [41,42,43], which may be subdivided into facets to allow a more refined analysis of a person’s personality [44].

The five personality traits listed in this concept are:openness to experience (inventive/curious vs. consistent/cautious)conscientiousness (efficient/organized vs. extravagant/careless)extraversion (outgoing/energetic vs. solitary/reserved)agreeableness (friendly/compassionate vs. critical/rational)neuroticism (sensitive/nervous vs. resilient/confident)

The number of facets to describe the personality traits of all individuals may probably be infinite; nevertheless, in various concepts on personality traits, researchers have had to establish definitions. Thus, different concepts for the description and measurement of personality traits have emerged in different areas of knowledge [42,45], namely, in the context of certain diseases [46,47]. On the basis of a personality trait profile, a prediction is possible about a person’s future behavior, whereby a distinction must be drawn between explicit personality traits, i.e., those accessible to the conscious mind, and implicit personality traits, which are not accessible to the conscious mind. Personality traits are generally stable over time [48] but can be influenced by significant changes in life conditions [49,50]. Explicit personality traits play an important role in research since they can be assessed, for example, by means of questionnaires, such as the “The Mini-IPIP” [51].

Personality traits have been widely investigated in the field of PD [52,53,54]. PwPs present a low novelty-seeking and high harm-avoidance profile in their personality traits [52]. The dopamine deficit found in PD may explain the low novelty-seeking profile, whereas the occurrence of affective disorders may explain the high harm-avoidance profile of PwPs [52]. PwPs’ personalities may have a positive or negative impact on the quality of life with PD [53]. Pontone et al. (2017) found that PwPs with otherwise similar disease burdens and depressive symptoms may experience different quality of life scores depending on the presence of neurotic or conscientious personality traits. Here, it becomes apparent that when assessing the PD-related quality of life and interpreting the data, it is important to differentiate whether these are induced by clinical symptoms of PD (e.g., depression, anxiety, motor impairments) and are treatable or reflect individual pre-morbid personality traits [53].

Consequently, personality traits should also be taken into account when considering coping styles in the co-design process, as it has been shown that personality traits such as neuroticism have an influence on an individual’s ability to assess and manage stressors and are, therefore, linked to coping styles.

Up to this point, we have been able to map why personality traits are important to care delivery in PD: personality traits generally remain stable in adulthood and can have an impact on the perceived quality of life of people with a chronic disease. Looking at a patient’s personality, e.g., by means of a questionnaire, at the beginning of his or her treatment, might therefore allow us to draw valuable insights into how the patient will eventually respond to future treatment plans.

In the next section, we will discuss why looking at a patient’s personality traits may not be enough to provide the best possible care delivery concept. We will take a closer look at how patients react to challenging situations and how this can be used advantageously in the development of novel integrated care concepts.

## 4. Coping Styles in PwPs: How Do They Influence the PwPs’ Journey?

The individual handling of a disease, in our case, PD, is defined by so-called coping styles. In general, the term “coping” or “coping strategy” refers to an individual’s approach to a life event or phase that is perceived as significant and difficult. Within the medical context, coping refers to the coping behaviors of people with chronic illnesses or disabilities. Coping has been described by Lazarus et al. (1984) as “the constant cognitive change and behavioral adaptation when handling specific external and/or internal demands that are evaluated as something that exceeds the resources of the person”. In this regard, according to Lazarus et al. (1984), coping can be considered a dynamic process as it consists of a series of reciprocal responses through which the individual and the environment interact and influence each other. Thus, strategies such as minimizing, avoiding, tolerating, and accepting a stressful situation can be carried out by the individual [36]. However, three categories for individual coping styles emerge: task orientation (taking useful steps to overcome or minimize the stressor promptly), emotion orientation (overcoming one’s own emotional stress response), and avoidant orientation (disengaging from the stressful situation or emotions) [55].

When faced with the same kind of stressors, individuals respond differently depending on their personality traits, social environments, individual life experiences, and available resources. In addition, over the course of life, coping styles are likely to remain unchanged even when living with a chronic progressive disease such as PD [56], whereas coping strategies may evolve [57,58]. Questionnaires to measure existing coping strategies in individuals, such as the COPE Inventory [59] or the Brief Resilient Coping Scale (BRCS) [60], are essential tools to assess the individual’s need regarding resources to cope with a given stressor and to identify appropriate strategies for the individual to overcome the stressor. An important criticism mentioned in research on coping styles emphasizes that by focusing on the style per se, the complexity and variability across the range of coping efforts are not adequately captured, thus affecting the interpretation of research findings. So, a focus on measuring coping styles within a specific context should improve the validity and quality of the results [36].

The individual’s way of coping with living with a chronic progressive illness, such as PD, has been well described in the scientific literature [61,62], to a similar extent as for personality traits. PwPs cannot recover from their diagnosis—there is no cure for PD. Thus, PwPs (and their care partners) simply are unable to “fix” their ongoing condition on their own; therefore, they must find strategies to adapt to the disease-related burden. Coping strategies are generally relevant in life, but they are especially important for people with chronic conditions. Among PwPs, the most common coping strategy employed is either action-oriented or problem-oriented coping [58,63,64]. There is also evidence that PwPs utilize all types of coping strategies to varying degrees. In this regard, Liebermann et al. (2020) found that some PwPs use more than one strategy at a time. The same authors call for future research to consider a combination of coping strategies in contrast to the established division between active and passive coping [65].

At this point, we would like to broaden our focus somewhat off the intrapersonal perspective to address two interpersonal coping perspectives that emphasize the importance of social relationships in the assessment of stressors and coping processes by examining the Systemic Transactional Model (STM) of dyadic coping [66,67] and communal coping [68,69]. The STM extends the model proposed by Lazarus and Folkman [36], suggesting that care partners engage in a primary assessment process in which they assess the importance of a situation to their own well-being, the well-being of their partner, and the well-being of the relationship as a whole [70]. Communal coping emphasizes the embeddedness of the individual in social relationships and the importance of interpersonal processes in coping with stressful experiences in life [68]. Partnerships and other social units (e.g., families, communities) are viewed as dynamic systems in which any change in one partner will naturally affect the other partner and thus influence the relationship as a whole [70]. Thus, communal coping may influence primary stress appraisal processes [69], and partners may perceive the stressor as a challenge or threat that is relevant to the couple [70]. Secondary stress appraisal processes are also likely influenced by communal coping, whereby the partner explicitly or implicitly draws on the partner’s available coping resources, additionally to their own, when evaluating their available coping resources. This results in the perceived doubling of available resources, which may make communal coping more effective in buffering stress than, for example, social support, where partners’ resources are available when needed but are still provided from one person to another rather than pooled or shared [68,70].

By calling for the consideration of coping styles in the co-design of integrated care concepts, we give an extra step and postulate that PwPs should be assessed not only according to their personality and coping strategies utilized but also their care partners [71]. It has been shown that multidisciplinary interventions aimed at improving the quality of life of PwPs may be more effective if informal care partners are also made aware of coping strategies and how they can play an active role in a positive psychosocial adjustment to the disease [72]. In addition, there is evidence that family members often function as informal caregivers and thus provide vital day-to-day support. Healthcare professionals and informal care partners need to know about coping strategies to better support patients [73].

## 5. Can the Inclusion of Coping Styles in the Co-Design of Integrated Care Concepts Be a Game Changer?

Summarizing the above, we can state that patients usually face stressors in a task-, emotion-, or avoidance-oriented manner. However, recent research has also indicated that this view may need to be broadened and that patients may engage in multiple coping strategies. This finding is particularly relevant in the development of new care concepts with the PwPs in a co-design approach. As an example, if only the needs and preferences of patients and care partners are taken into account and coping styles for a patient do not usually adopt avoidance-oriented coping styles towards the experience of PD, care concepts that are highly task-oriented will have limited or no utility. The effectiveness and acceptance of care delivery by PwPs and care partners might rise if new integrated care concepts are adapted to the coping styles of PwPs. Like personality traits, the early assessment of a patient’s coping style may help in the design of care plans using standardized questionnaires.

Consequently, it is necessary to conduct research on the incorporation of personality traits and coping styles in the co-design process of integrated care concepts and how these concepts can contribute to the enhanced tailoring of integrated care to the actual needs of PwPs and their care partners and thus address prevailing problems in healthcare systems, such as the fragmentation of health care services and the scarcity of resources.

Personality traits and coping styles may be considered in the co-design process of integrated care concepts in the following manner in Table 2.

In this opinion paper, we have highlighted the research on personality traits and coping styles and the benefits of their adoption when delivering care to PwPs. Currently, there is no scientific research on the inclusion of individual coping styles of PwPs in the design process of integrated care concepts. We provide an initial insight on how the consideration of coping styles may be beneficial not only for the PwP but also for their care partner and care delivery as a whole. It is time to involve PwPs as partners in the design process of integrated care concepts.

## Figures and Tables

**Table 1 jpm-12-00921-t001:** Phases of experience-based co-design (EBCD).

Phase of EBCD *	Proposed Action for the Participants *
(1) Set-up	Setting up the project (administration, project management arrangements)
(2) Staff Engagement	Gather experiences of staff with observational fieldwork and in-depth interviews Review findings with staff and prioritize main findings to improve services
(3) Patient Engagement	Gather experiences of patients and care partners through narrative interviews Review findings with patients and care partners and prioritize main findings to improve services
(4) Connecting stakeholders and exchange	Connect all stakeholders Share experiences in an initial co-design event Identify priorities for change
(5) Co-design activities	Co-work in small groups Focus on identified priorities Design and implement improvements to services
(6) Review and renewal	Collaborative assessment by participants

* Following Bate and Robert, 2007; Robert, 2013; Donetto et al. 2015; own illustration.

**Table 2 jpm-12-00921-t002:** Proposed framework to adopt personality traits and coping styles in the development of integrated care models.

Promotion	Point in Time of Co-Design Process *	Type of Research	Instrument **
Assessment of personality traits PwP Care partner	(1) Set-up	Literature review of tools appropriate to setting and stakeholder	_
	(3) Patient Engagement	Quantitative evaluation	PwPs and Care Partners: The Revised Neo Personality Inventory (NEO PI-R) [74], Minnesota Multiphasic Personality Inventory-2 (MMPI-2) [75]
Assessment of coping strategy PwP Care partner	(1) Set-up	Literature review of tools appropriate to setting and stakeholder	_
	(3) Patient Engagement	Quantitative evaluation	PwPs: COPE inventory [59] or Brief COPE [76]; The Dyadic Coping Inventory (DCI) [77] Care partner: The Dyadic Coping Inventory (DCI) [77]
Staff Education — increase knowledge — build acceptance for considering personality traits and coping styles in care delivery	(2) Staff Engagement	Training/Workshop (online or face-to-face)	_
Patient education — reflection of results of the previous assessments — increase knowledge — increase awareness of the relevance of coping styles in co-design — provide potential approaches on how knowledge can be useful for patients	(3) Patient Engagement	Training/Workshop (online or face-to-face)	_
Care partner education — reflection of results of the previous assessments — increase knowledge — increase awareness of the importance of coping styles in co-design — provide potential approaches on how knowledge can be useful for the patients/care partner personally — increase awareness that personality traits and coping styles of care partner are important	(3) Patient Engagement	Training/Workshop (online or face-to-face)	_
Exchange and mutual discussion: — personality traits and coping styles — build mutual understanding — emphasize the importance of consideration	(4) Connecting stakeholders and exchange	Team Discussion	_
Continuous review: — ensure personality traits and coping styles are appropriately reflected — throughout the whole process	(1)–(6)	Team Discussion	_

* Following the exemplified EBCD process; see Table 1. ** The list of tools given here serves solely as an example as there are numerous tools available.

## Data Availability

Not applicable.

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
