# Peer review of "Coping Styles in Patients with Parkinson’s Disease: Consideration in the Co-Designing of Integrated Care Concepts"

_jpm, 2022, doi:10.3390/jpm12060921_

Round 1

Reviewer 1 Report

Parkinson’s disease (PD) is the second most common degenerative disorder after Alzheimer disease. In PD are not only clinical motor (rigidity, bradykinesia, tremor), but also a wide range of clinical non-motor symptoms (some psychiatric like depression and anxiety) that may be addressed in individual way for PD patient.

This is an original paper that draw attention of the importance of early evaluating the personality traits and their response strategies to stress of people living with Parkinson’s (PwPs) in order to elaborate a patient-centered plan for the patients and their partner to cope with PD.

PD patients usually have a low novelty-seeking and a high harm-avoidance profile in their personality traits.

The authors emphasizes the co-design approaches – patients, family members, doctors – working together in order to find the best care plan for each PD patient and they  propose a framework of how to use personality traits and coping styles in the development of integrated care models.

Author Response

Thank you for taking time to review and comment the manuscript – we appreciate that our efforts have been well received.

Reviewer 2 Report

Although the objective and topic of the study are interesting and impactful, the study is not well structured from a practical point of view. The clinical aspect of assistance and the methods of intervention are not clear. I propose to structure the article with clearer sections of materials, methods and conclusions.
The  research design isn't clear

Author Response

We thank Reviewer 2 very much for his/her comments on our manuscript. We appreciate the feedback and agree with the reviewer's assessment.

Unfortunately, we have to admit that in the final formatting before submission of the paper a mistake was made: the paper was written as an 'opinion paper' and not as a 'concept paper'. For this reason, none of the traditional divisions of a research paper into 'Introduction/Background', 'Methodology', 'Results', 'Discussion' and 'Conclusion' were chosen. We would like to thank reviewer 2 for noticing this mistake and from now on we will make sure to choose the appropriate category when submitting the paper. Therefore, this has now been corrected in the manuscript and made transparent.

We are therefore confident that Reviewer 2's comments on the research design, the presentation of results, and the conclusions are now no longer applicable. If this is not the case, we will gladly accept further suggestions for changes.

Reviewer 2 further points out that especially the clinical aspect of support and the method of intervention is not clarified in our manuscript. We would like to point out anew at this point that this is an opinion paper that, in its scope, provides for the first time an important, but of course mimetic, perspective on the topic presented. In Table 2 of the manuscript, we have attempted to address how to potentially integrate our demand for the consideration of personality traits and coping styles by proposing a framework whose phases follow the established process of co-design. Here we exemplify some of the possible ways and means of implementation from the point of view of both PwPs and care partners as well as health care professionals into the co-design process.

We hope that we were able to fully engage Reviewer 2 with our work and would like to express our thanks once again for the valuable comments.

Reviewer 3 Report

Very nice perspective. I have no comments

Author Response

We thank Reviewer 3 very much for the effort you have made and are pleased that our work has been well received.